# Robust Differential Abundance Analysis of Microbiome Sequencing Data

**DOI:** 10.3390/genes14112000

**Published:** 2023-10-26

**Authors:** Guanxun Li, Lu Yang, Jun Chen, Xianyang Zhang

**Affiliations:** 1Department of Statistics, Texas A&M University, College Station, TX 77843, USA; guanxun@stat.tamu.edu; 2Department of Quantitative Health Sciences, Mayo Clinic, Rochester, MN 55905, USA; yang.lu@mayo.edu

**Keywords:** compositional data, differential abundance analysis, Huber regression, robustness, winsorization

## Abstract

It is well known that the microbiome data are ridden with outliers and have heavy distribution tails, but the impact of outliers and heavy-tailedness has yet to be examined systematically. This paper investigates the impact of outliers and heavy-tailedness on differential abundance analysis (DAA) using the linear models for the differential abundance analysis (LinDA) method and proposes effective strategies to mitigate their influence. The presence of outliers and heavy-tailedness can significantly decrease the power of LinDA. We investigate various techniques to address outliers and heavy-tailedness, including generalizing LinDA into a more flexible framework that allows for the use of robust regression and winsorizing the data before applying LinDA. Our extensive numerical experiments and real-data analyses demonstrate that robust Huber regression has overall the best performance in addressing outliers and heavy-tailedness.

## 1. Introduction

The human microbiome is a complex and multifaceted ecosystem comprising diverse microorganisms, including bacteria, viruses, and fungi. These microorganisms inhabit different parts of the human body and play a crucial role in various biological functions essential for human health and disease prevention [1,2]. Understanding the diversity and abundance of microorganisms in the microbiome, such as the gut microbiome, is crucial for identifying potential pathogens that can cause harm or probiotics that promote good health [3,4].

Metagenomic sequencing is the primary method used to study the microbiome. It provides a comprehensive snapshot of the microbiome’s composition by sequencing the genetic material of all microorganisms in a sample [5,6]. However, this technique only provides relative abundance data, with the abundance of each microorganism expressed as a proportion of the total number of microorganisms in the sample [7]. Absolute abundance measurement can be achieved through various experimental techniques like qPCR, spike-in, and flow cytometry. However, these techniques have yet to be widely adopted due to their severe limitations [8]. Therefore, the prevailing sequencing protocol is still only capable of measuring the relative abundances. Nevertheless, when combined with appropriate statistical methods, relative abundance data can still provide valuable insights into the composition and function of the microbiome and its impact on health and disease. Following the processing of sequence reads using a bioinformatic pipeline, such as DADA2 [9] for 16S-targeted sequencing and MetaPhlAn2 [6] for shotgun metagenomic data, an abundance table that records the frequencies of the detected microbial taxa is generated. This table is used for downstream statistical analyses along with metadata that captures sample-level characteristics.

Differential abundance analysis (DAA) is a central downstream task that seeks to identify microbial taxa whose abundance correlates with a variable of interest. However, classical statistical tools, such as ANOVA and rank-based tests, are unsuitable for DAA as they do not account for the compositional nature of microbiome data and can lead to significant false discoveries. This is due to changes in one microorganism’s abundance that can influence others’ relative abundance. Several differential analysis methods have been developed that consider compositional effects [10] to address this issue. These methods use either robust normalization techniques, such as TMM, RLE, CSS, and GMPR [11,12,13,14] or (log) ratio-based approaches, including ALDEx2 [15], ANCOM-BC [16], MaAsLin2 [17], and LinDA [18].

Identifying differentially abundant taxa in real-world data can be challenging due to the presence of outliers (overly abundant taxa) that may negatively impact the effectiveness of existing DAA methods [19,20,21], where outliers mean an extremely high abundance of a particular taxon in a few samples. Despite their widespread use, there has been a need for more systematic investigations into the influence of outliers on DAA methods. Additionally, log-ratio-based approaches (e.g., LinDA, ANCOM-BC, MaAsLin2) rely on normality assumption for random errors, which may not be valid for real-world data. Our observations suggest that this assumption can be violated in real-world data, leading to a phenomenon we term heavy-tailedness, which can impair the performance of existing DAA methods. Here, heavy-tailedness means that the tail of the distribution of random errors is heavier than the normal distribution.

This study aims to investigate the impact of outliers and heavy-tailedness on differential abundance analysis (DAA) methods and identify effective strategies to mitigate these challenges. LinDA was used as the benchmark DAA method, and the primary aim was to identify the most effective strategy to address outliers and heavy-tailedness in DAA. The study revealed that the presence of outliers or heavy-tailedness significantly reduces the power of detecting differential taxa. To address these challenges, we proposed a general M-estimation framework for DAA, which encompasses differential analysis based on Huber regression as a special case. Huber regression is a widely used statistical method that guards against outliers and heavy-tailedness in regression problems [22]. Additionally, this study explored the effectiveness of the winsorization method, a statistical data pre-processing technique in combination with LinDA, for handling noisy data. Winsorization replaces extreme values with less extreme ones and has been shown to effectively handle outliers and heavy-tailedness in data analysis [23]. This study’s simulations demonstrated that Huber regression exhibits superior robustness against outliers and heavy-tailedness compared to the LinDA and LinDA methods with winsorization. Therefore, this study recommends using the Huber method in instances wherein the dataset is subject to noise.

We summarize our main contributions as follows:1.This study conducted comprehensive simulations to investigate the impact of outliers and different types of heavy-tailedness on the various DAA methods.2.This study introduced a general M-estimation framework for DAA, which includes several methods that are robust to outliers and heavy-tailedness.3.This study conducted extensive simulations to examine the performance of various DAA methods to address outliers and heavy-tailedness, including multiple levels of the winsorization technique and different M-estimation-based methods. The experiments demonstrated that the proposed Huber regression method based on the M-estimation framework is more stable for outliers and heavy-tailedness.

The remainder of this paper is organized as follows. In Section 2.1, we introduce the regression-based framework for DAA, which includes LinDA and the Huber regression-based method as special cases. Section 2.2 provides an overview of the winsorization method and its implementation details. Our simulation studies are presented in Section 3. Lastly, Section 4 presents some real data analyses. Additional numerical results can be found in the Appendix A, Appendix B and Appendix C.

## 2. Materials and Methods

### 2.1. A Regression-Based Framework for Differential Analysis

In this section, we present a generalization of the LinDA method, utilizing a M-estimation framework for DAA based on the central log ratio (CLR) transformation. By replacing the $l_{2}$ loss with a robust loss function, we propose a more robust approach for DAA.

#### 2.1.1. CLR-Based Log-Linear Models

Let $X_{is}$ and $Y_{is}$ denote the absolute abundance and observed read count of the *i*-th taxon in the *s*-th sample, respectively. For the *s*-th sample, the total read count of all taxon, $N_{s}=\sum_{i=1}^{m}Y_{is}$, is determined by the sequencing depth and DNA materials. Given $N_{s}$, it is natural to model the stratified count data over *m* taxon using a multinomial distribution as
(1)$$P\left(Y_{1s}=y_{1s},\ldots ,Y_{ms}=y_{ms}\right)=\frac{N_{s}!}{\prod_{i=1}^{m}y_{is}!}\prod_{j=1}^{m}{\left(\frac{X_{js}}{\sum_{i=1}^{m}X_{is}}\right)}^{y_{js}}.$$
Under (Equation 1), we have
(2)$$\log\left(\frac{Y_{is}}{\sum_{j=1}^{m}Y_{js}}\right)=\log\left(\frac{X_{is}}{\sum_{j=1}^{m}X_{js}}\right)+e_{is},$$
where the ratio on the left side represents the sample proportion, while the ratio on the right side represents the population proportion, and $e_{is}$ denotes the estimation error. The sample proportion is the maximum likelihood estimator (MLE) of the population proportion based on the multinomial model (1). By the consistency of the MLE, $e_{is}$ is expected to diminish as $N_{s}$ increases. We consider the log linear model on the absolute abundance
(3)$$\log\left(X_{is}\right)=u_{s}\alpha_{0i}+c_{s}^{⊤}\beta_{0i}+\epsilon_{is},$$
where $c_{s}={\left(1,c_{s1},\ldots ,c_{sd}\right)}^{⊤}$ includes intercept and the *d*-dimensional covariates to be adjusted, $u_{s}$ is the variable of interest, and $\epsilon_{is}$ is the error term. Here, we assume a log-linear model on the absolute abundance, which is a reasonable and widely adopted approach in abundance data analysis [24,25]. The objective here is to identify taxa that show differential abundance relative to $u_{s}$. To this end, we simultaneously test the following *m* hypotheses:$$H_{0,i}:\alpha_{0i}=0\;\mathrm{versus}\;H_{a,i}:\alpha_{0i}\neq 0,\;1\leq i\leq m.$$
Set $\varepsilon_{is}=\epsilon_{is}+e_{is}$. Under (Equation 2) and (Equation 3), the CLR data satisfy the following linear model: $$\begin{array}{cc} W_{is}:= & \log\left\{\frac{Y_{is}}{{\left(\prod_{j=1}^{m}Y_{js}\right)}^{1/m}}\right\}=\log\left(\frac{Y_{is}}{\sum_{k=1}^{m}Y_{ks}}\right)-\frac{1}{m}\sum_{j=1}^{m}\log\left(\frac{Y_{js}}{\sum_{k=1}^{m}Y_{ks}}\right)\\ = & \log\left(X_{is}\right)-\frac{1}{m}\sum_{j=1}^{m}\log\left(X_{js}\right)+e_{is}-\frac{1}{m}\sum_{j=1}^{m}e_{js}\\ = & u_{s}\left(\alpha_{0i}-\bar{\alpha }\right)+c_{s}^{⊤}\left(\beta_{0i}-\bar{\beta }\right)+\varepsilon_{is}-{\bar{\varepsilon }}_{s}, \end{array}$$
where $\bar{\alpha }=m^{-1}\sum_{i=1}^{m}\alpha_{0i}$, $\bar{\beta }=m^{-1}\sum_{i=1}^{m}\beta_{0i}$, and ${\bar{\varepsilon }}_{s}=m^{-1}\sum_{i=1}^{m}\varepsilon_{is}$.

#### 2.1.2. M-Estimation Framework for Differential Analysis

Define ${\bar{\alpha }}_{i}=\alpha_{0i}-\bar{\alpha }$ and ${\bar{\beta }}_{i}=\beta_{0i}-\bar{\beta }$. We propose to estimate ${\bar{\alpha }}_{i}$ and ${\bar{\beta }}_{i}$ by solving the following M-estimation problem:(4)$$\left({\tilde{\alpha }}_{i},{\tilde{\beta }}_{i}\right)=\underset{\alpha_{i},\beta_{i}}{\arg\;\min}\frac{1}{n}\sum_{s=1}^{n}L\left(W_{is}-u_{s}\alpha_{i}-c_{s}^{⊤}\beta_{i}\right),$$
where L is a loss function chosen by the practitioners and the resulting estimators are referred to as M-estimators. When L is the $l_{2}$ loss, i.e., $L\left(r\right)=r^{2}$, $\left({\tilde{\alpha }}_{i},{\tilde{\beta }}_{i}\right)$ become the ordinary least squares (OLS) estimators, which has been considered in [18].

**Example** **1.**
*To handle heavy-tailedness and outliers in datasets, we can employ the robust loss function in (Equation 4), which is commonly used in robust regression. Robust regression is designed to estimate the parameters of a regression model in the presence of influential observations that can distort the results. Unlike ordinary least squares regression, which assumes normally distributed errors with constant variance, robust regression offers more flexible assumptions about the error distribution and is less sensitive to outliers. It becomes particularly advantageous when data contain outliers that cannot be readily removed or explained, or when ensuring that the regression coefficients are not overly influenced by a few observations. Below are some popular robust regression loss functions.*
1.
*Huber’s loss is defined as*

$$l_{\mathrm{Huber}}\left(r\right)=\left\{\begin{array}{cc} \frac{1}{2}r^{2} & \mathrm{if}\;|r|\leq c\\ c|r|-\frac{1}{2}c^{2} & \mathrm{if}\;|r|>c \end{array}\right.,$$


*where c is the hyperparameter and the default value is 1.345, which is widely used in robust regression studies. Notice that the Huber estimator down-weights the influence of observations with large residuals, resulting in less impact on the estimated regression coefficients. Additionally, Huber [22] argued that if the true distribution was normal, this loss function is asymptotically 95% as efficient as least squares.*
2.
*Tukey’s bisquare loss is defined as*

$$l_{\mathrm{bi}}\left(r\right)=\left\{\begin{array}{cc} \frac{c_{0}^{2}}{6}\left(1-{\left(1-{\left(\frac{r}{c_{0}}\right)}^{2}\right)}^{3}\right) & \mathrm{if}\;|r|\leq c_{0}\\ \frac{c_{0}^{2}}{6} & \mathrm{if}\;|r|>c_{0} \end{array}\right.,$$


*where $c_{0}=4.685$ is the standard constant for this loss function. It is worth noting that this function has been shown to possess an asymptotic efficiency of 95% with respect to linear regression for the normal distribution.*
3.
*Quantile regression loss is defined as*

$$l_{\tau }\left(r\right)=\left\{\begin{array}{cc} \tau r & \mathrm{if}\;r\geq 0\\ (\tau -1)r & \mathrm{if}\;r<0 \end{array}\right.,$$


*where τ represents the quantile level of interest. Notably, when $\tau =\frac{1}{2}$, the loss function is equivalent to the $l_{1}$ loss, expressed as $l_{1}\left(r\right)=\left|r\right|$. The $l_{1}$ loss corresponds to the loss function utilized in the least absolute deviations (LAD) regression method.*



Let $z_{s}={\left(u_{s},c_{s}^{⊤}\right)}^{⊤}$ and $\theta_{i}={\left(\alpha_{i},\beta_{i}^{⊤}\right)}^{⊤}$. We define ${\tilde{\theta }}_{i}$ (and ${\bar{\theta }}_{i}$) in the same way as $\theta_{i}$ by replacing $\alpha_{i}$ with ${\tilde{\alpha }}_{i}$ (and ${\bar{\alpha }}_{i}$), and $\beta_{i}$ with ${\tilde{\beta }}_{i}$ (and ${\bar{\beta }}_{i}$), respectively. Denote $r_{is}=W_{is}-z_{s}^{⊤}\theta_{i}$, and define ${\bar{r}}_{is}$ and ${\tilde{r}}_{is}$ analogously by replacing $\theta_{i}$ with ${\bar{\theta }}_{i}$ and ${\tilde{\theta }}_{i}$, respectively. We can then rewrite each summand of the objective function in (Equation 4) as $L\left(r_{is}\right):=L\left(W_{is}-z_{s}^{⊤}\theta_{i}\right)$. Observe that
$$\frac{\partial }{\partial \theta_{i}}L\left(r_{is}\right)=z_{s}\psi \left(r_{is}\right),$$
where ψ is referred to as the influence curve [26]. Under certain regularity conditions (see, for example, Theorem 5.21 in Van der Vaart [27]), the asymptotic normality of the M-estimator is given as follows [26]:(5)$$\sqrt{n}\left({\tilde{\theta }}_{i}-{\bar{\theta }}_{i}\right)\to^{d}N\left(0,\Sigma_{i}\right),$$
where
(6)$$\Sigma_{i}={\left(\frac{1}{n}\sum_{s=1}^{n}z_{s}z_{s}^{⊤}\right)}^{-1}\frac{E[\psi^{2}\left({\bar{r}}_{is}\right)]}{E{\left[\psi^{\prime }\left({\bar{r}}_{is}\right)\right]}^{2}}.$$
In practice, we will not know the true distribution of the error term and the true regression parameters ${\bar{\theta }}_{i}$. Therefore, we propose using the plug-in method to estimate the asymptotic variance of ${\tilde{\theta }}_{i}$ (rather than $\sqrt{n}{\tilde{\theta }}_{i}$), which is given by
(7)$${\hat{\Sigma }}_{i}={\left(\sum_{s=1}^{n}z_{s}z_{s}^{⊤}\right)}^{-1}\frac{\frac{1}{n}\sum_{s=1}^{n}\psi^{2}\left({\tilde{r}}_{is}\right)}{{\left(\frac{1}{n}\sum_{s=1}^{n}\psi^{\prime }\left({\tilde{r}}_{is}\right)\right)}^{2}}.$$
When L is the $l_{2}$ loss, the estimators used in LinDA and (Equation 7) are asymptotically equivalent.

From the above discussions, ${\tilde{\alpha }}_{i}$ obtained by minimizing (Equation 4) is an asymptotically unbiased estimator for $\alpha_{i}-\bar{\alpha }$. To estimate $\alpha_{i}$, it remains to find an estimator for $\bar{\alpha }$. To this end, we adopt the mode correction method proposed by Zhou et al. [18]. Specifically, assuming that only a small portion of taxa exhibit differential abundance, meaning that most values of $\alpha_{i}$ are equal to 0. Under this assumption, the mode of ${\tilde{\alpha }}_{i}$ is expected to be close to $-\bar{\alpha }$. Hence, we can estimate $\bar{\alpha }$ through estimating the mode of $\left\{{\tilde{\alpha }}_{i}:1\leq i\leq m\right\}$. Specifically, we use the kernel smoothing approach to estimate the mode and let
$$\tilde{\alpha }:=-\frac{\hat{\mathrm{mode}}\left({\left\{\sqrt{n}{\tilde{\alpha }}_{i}\right\}}_{i=1}^{m}\right)}{\sqrt{n}}$$
be the estimate for $\bar{\alpha }$. Here,
$$\hat{\mathrm{mode}}\left({\left\{\sqrt{n}{\tilde{\alpha }}_{i}\right\}}_{i=1}^{m}\right)=\underset{a\in R}{\arg\;\max}\frac{1}{mh}\sum_{i=1}^{m}K\left(\frac{a-\sqrt{n}{\tilde{\alpha }}_{i}}{h}\right),$$
where K is a kernel function satisfying $\int_{-\infty }^{\infty }K\left(x\right)\;dx=1$ and *h* is the bandwidth. Then, the resulting bias-corrected estimator of $\alpha_{i}$ is given by ${\hat{\alpha }}_{i}={\tilde{\alpha }}_{i}+\tilde{\alpha }$. In our implementation, we use *mlv* function in R package *modeest* to estimate the mode.

Once the bias-corrected estimator is obtained, we implement the *t*-test by defining the test statistic as $T_{i}={\hat{\alpha }}_{i}/{\hat{\sigma }}_{i}$, where ${\hat{\sigma }}_{i}^{2}$ is the variance estimator of ${\tilde{\alpha }}_{i}$ in the regression problem, corresponding to the (1,1)th entry of ${\hat{\Sigma }}_{i}$ defined in (Equation 7). Although we have the asymptotic normality of the M-estimator, our simulations revealed that the *t*-distribution provides a better approximation for the sampling distribution of $T_{i}$ and offers better FDR control for small samples. Consequently, we propose using the *t*-test in our method. The *p*-value is then calculated as $2P(T\geq |T_{i}|)$, where *T* follows a *t*-distribution with n−d−2 degrees of freedom. To control the FDR, we recommend using the Benjamini–Hochberg (BH) procedure to adjust the *p*-values obtained for each taxon. The taxa with an adjusted *p*-value less than a certain threshold are referred to as differential taxa.

We summarize our procedure as follows:1.For each taxon, solve the optimization problem defined in (Equation 4) to obtain the estimator ${\tilde{\alpha }}_{i}$ and its variance estimator ${\hat{\sigma }}_{i}^{2}$.2.Calculate the mode based on $\left\{{\tilde{\alpha }}_{i}:1\leq i\leq m\right\}$, and perform mode correction to obtain the bias-corrected estimator ${\hat{\alpha }}_{i}$.3.Compute the test statistics $T_{i}$ and the *p*-value for each taxon.4.Apply the BH procedure to adjust the *p*-values.

**Remark** **1.**
*Given the practical challenge of determining the optimal hyperparameter to use in the loss function (e.g., c in the Huber loss function), we propose utilizing the Cauchy combination rule [28] for aggregating the p-values obtained from different hyperparameters. This approach addresses the difficulty of selecting the most suitable hyperparameter by combining the results from multiple options.*


**Remark** **2.**
*Based on the regression-based framework, our method can be readily extended to apply to the mixed-effect model. Please refer to Appendix B for more details.*


### 2.2. Winsorization

Winsorization is a statistical data preprocessing technique utilized for handling outliers and heavy-tailed data. This method involves sorting the dataset in either ascending or descending order based on the analytical requirements. Next, the extreme values, namely outliers or values in the tails of the distribution, are substituted with the smallest or largest non-outlier value, correspondingly. Moreover, winsorization can be complemented with DAA techniques to enhance the precision and robustness of statistical analysis, particularly when confronted with datasets containing outliers or heavy-tailed distributions.

In microbiome data analysis, winsorization typically entails replacing the top 1−τ largest values with the value at the τ quantile [20]. Specifically, for a given taxon *i*, $Y_{is}$ was arranged in increasing order, and the τ quantile of ${\left\{Y_{is}\right\}}_{s=1}^{n}$ was computed and denoted as $q_{i}\left(\tau \right)$. We replace the observed count $Y_{is}$ by its winsorized value defined as
$$\begin{array}{c} {\tilde{Y}}_{is}=\left\{\begin{array}{cc} Y_{is} & \mathrm{if}\;Y_{is}\leq q_{i}\left(\tau \right),\\ q_{i}\left(\tau \right) & \mathrm{if}\;Y_{is}>q_{i}\left(\tau \right). \end{array}\right. \end{array}$$

While winsorization is known to reduce the influence of outliers, it remains unclear whether this method leads to the loss of essential data information. Furthermore, despite its widespread use, the impact of winsorization on DAA methods has not been thoroughly investigated. Consequently, this study seeks to comprehensively examine the impact of multiple levels of winsorization on DAA methods in various scenarios, both with and without outliers/heavy-tailedness, through several numerical examples.

## 3. Results

To comprehensively evaluate the performance of different M-estimation-based methods and winsorization at different levels, we conducted extensive simulations under various scenarios. Before describing the simulation settings, we provide a list of the methods we compared.

We evaluated six methods in our comparative analysis, namely LinDA [18] without winsorization (referred to as LinDA), LinDA with winsorization at the 97% quantile (referred to as LinDA97), LinDA with winsorization at the 90% quantile (referred to as LinDA90), M-estimation method with Huber’s loss (referred to as Huber), M-estimation method with Tukey’s bisquare loss (referred to as Bi_square), and quantile regression method (referred to as QR). To implement the Huber and bisquare methods, we used the *rlm* function in the *MASS* package (version: 7.3-60) in R to perform the regression estimation. For the selection of hyperparameters, we considered 10 values of *c* equally spaced within the interval [1.345,5] on a log scale for the Huber method. For the Bi_square method, we took 10 values of $c_{0}$ equally spaced within the interval [4.685,20] on a log scale. We used the *rq* function in the *quantreg* package (version: 5.95) in R to implement quantile regression. Similar to the Huber method, we took into account the quantile level τ that is evenly distributed across the interval [0.25,0.75] with an adjacent difference of 0.05 (resulting in a total of 11 values).

**Remark** **3.**
*We also compared with other differential abundance analysis methods, including ALDEx2 [15], ANCOM-BC [16], and MaAsLin2 [17], and the results are deferred to Appendix C. Based on our simulation, we found that LinDA outperforms other methods that do not consider outliers and heavy-tailedness. Hence, we used LinDA as the benchmark method in the following and focus on comparing various log-linear model-based approaches used for addressing outliers.*


To handle zero values, we employed the hybrid method proposed by Zhou et al. [18], which combines two different approaches. The first approach involves adding a pseudo-count of 0.5 to all counts, which is a commonly used technique in microbiome data analysis on the log scale. The second approach is the imputation-based method, which involves replacing the zeros with fractions equal to $N_{s}/\max\left\{N_{k}:Y_{ik}=0\right\}$ for the *i*-th taxon in the *s*-th sample, where larger fractions are used for samples with larger library sizes. Zhou et al. [18] used a statistical test to determine which method to apply. Specifically, they tested the association between the covariate of interest and the library size based on the log-linear model. If the *p*-value was less than 0.1, they used the imputation approach; otherwise, they used the pseudo-count approach. More details can be found in Zhou et al. [18].

### 3.1. Simulations Based on Log-Linear Models

In this section, we simulated datasets from log-linear models. We adopted the data-generating process proposed by Zhou et al. [18], while using different methods to generate error terms. We assumed that the baseline absolute abundance $X_{is}^{*}$ is generated from
$$\log\left(X_{is}^{*}\right){∼}^{i.i.d.}N\left(\beta_{i}^{*},\sigma_{i*}^{2}\right),$$
and the true proportion is obtained as
$$\pi_{is}^{*}=\frac{X_{is}^{*}}{\sum_{j=1}^{m}X_{js}^{*}}.$$
Letting ${\bar{\pi }}_{i}^{*}=\sum_{s=1}^{n}\pi_{is}^{*}/n$. We denote the signal strength as μ. In order to construct a power curve, we included six signal strengths in the figures, which are evenly spaced within the interval [1.05,2]. Given that low-abundance taxa exhibit lower statistical power, we assigned greater weight to their effects to prevent dominance by the abundant ones. Specifically, for the *i*-th taxon, we set
$$\mu_{i}=\log\left(2\mu \right)1\left({\bar{\pi }}_{i}^{*}>5\times 10^{-3}\right)+\log\left(2\mu {\left(5\times 10^{-3}/{\bar{\pi }}_{i}^{*}\right)}^{1/3}\right)1\left({\bar{\pi }}_{i}^{*}\leq 5\times 10^{-3}\right)$$
for n=50 and
$$\mu_{i}=\log\left(\mu \right)1\left({\bar{\pi }}_{i}^{*}>5\times 10^{-3}\right)+\log\left(\mu {\left(5\times 10^{-3}/{\bar{\pi }}_{i}^{*}\right)}^{1/3}\right)1\left({\bar{\pi }}_{i}^{*}\leq 5\times 10^{-3}\right)$$
for n=200.

We randomly selected the differential taxon from the entire set and denoted $\gamma_{i}$ as an indicator of whether the taxon is differentially abundant ($\gamma_{i}=1$) or not ($\gamma_{i}=0$). The underlying truth of $\gamma_{i}$ was generated from a Bernoulli distribution with parameter $p_{\gamma }$, where we set $p_{\gamma }=0.05$ or $p_{\gamma }=0.2$ to correspond to sparse and dense signal settings, respectively. We then denoted the true signal strength for the differentially abundant taxon by $\alpha_{i}=\mu_{i}\gamma_{i}$.

To generate the absolute abundance $X_{is}$, we considered two cases: with or without confounders, and three types of error terms. We defined $\epsilon_{is}$ as the error term corresponding to the *s*-th sample of the *i*-th taxon, which we will define later. The absolute abundance $X_{is}$ was then obtained by
$$\log\left(X_{is}\right)=\left\{\begin{array}{cc} \beta_{i0}+u_{s}\alpha_{i}+\epsilon_{is} & \mathrm{without}\;\mathrm{confounders},\\ \beta_{i0}+u_{s}\alpha_{i}+c_{s}^{⊤}\beta_{i}+\epsilon_{is} & \mathrm{with}\;\mathrm{confounders}, \end{array}\right.$$
where $\beta_{i0}$ is the intercept term and
$$u_{s}∼\mathrm{Bernoulli}\left(0.5\right)$$
if there is no confounder and
$$u_{s}∼\mathrm{Bernoulli}\left(1/\left(1+\exp(-0.5c_{s1}-0.5c_{s2})\right)\right)$$
if there are confounders. Here, the confounders $c_{s}=\left(c_{s1},c_{s2}\right)\in R^{n\times 2}$, with $c_{s1}∼\mathrm{Bernoulli}\left(0.5\right)$ and $c_{s2}∼N\left(0,1\right)$. The taxon-specific coefficients $\beta_{i1}$ and $\beta_{i2}$ are independently generated from the normal distributions with means of 1 and 2 and variances of 1, respectively. Then, the observed operational taxonomic unit (OTU) data were generated from
$$\left(Y_{1s},\cdots ,Y_{ms}\right)∼\mathrm{Multinomial}\left(N_{s},\pi_{1s},\cdots ,\pi_{ms}\right),$$
where $\pi_{is}=X_{is}/\sum_{j=1}^{m}X_{js}$. Moreover, the parameters $\beta_{i}^{*}$, $\sigma_{i*}^{2}$, and $N_{s}$ used in our study are identical to those used in Zhou et al. [18]. Specifically, Zhou et al. [18] estimated the aforementioned parameters based on a real dataset (COMBO) that studied the gut microbiota in a general population.

We begin by examining the performance of the methods in the absence of heavy-tailedness and outliers. We aim to assess how much power is lost when using robust regression/winsorization. For this purpose, we sampled errors from a normal distribution with a mean of 0 and a variance of $\sigma_{i*}^{2}$. The results are presented in Section A.1. We consider sample sizes of n∈{50,200}, referred to as the small sample case and the large sample case, respectively. We set the number of taxa as m=500. In the small sample size scenario, LinDA97 shows the best performance in the sparse-signal setting, whereas LinDA outperforms other methods in the dense-signal setting. For larger sample sizes, LinDA, LinDA97, Huber, and Bi_square methods all perform similarly and outperform LinDA90. The QR method has the lowest power in all scenarios and this method has FDR inflation when n=200 in the sparse-signal setting.

#### 3.1.1. Heavy-Tailedness Setting

To demonstrate the advantages of our method in handling heavy-tailedness, instead of assuming a standard normal distribution for $\epsilon_{is}$, we generate error terms using three different ways:Case 1: Student’s *t*-distribution with degrees of freedom 3.Case 2: Log-normal distribution with log mean parameter of 0 and log standard deviation parameter of 0.8. We recentered the samples so that it has a zero mean.Case 3: Weibull distribution with shape parameter of 0.5 and scale parameter of 0.3. We recentered the samples so that it has a zero mean.

These three are all heavy-tailed distributions, also discussed in Fan et al. [29]. It is important to underscore the significance of choosing appropriate parameters for generating error terms. For example, when the shape parameter of the Weibull distribution is small (e.g., 0.25), the resulting data can become excessively noisy, causing all methods to have no power to detect differential taxa. On the other hand, when the shape parameter of the Weibull distribution exceeds 1 (meaning the Weibull distribution is no longer heavy-tailed), the noise level diminishes significantly. As a result, all methods tend to achieve a power nearing 1.

We consider two sample size scenarios: n∈{50,200}, denoted as the small and large sample cases, respectively. Additionally, we fix the number of taxa at m=500. We conducted 100 simulation runs for each setting, calculated the mean power to detect differential taxa, and computed the empirical FDR. The results are presented in plots. This section showcases results for settings without confounders in the sparse scenario ($p_{\gamma }=0.05$). Similar phenomena were observed in both the dense scenario and the case with confounders. These are detailed in Section A.2 and Section A.3, respectively.

Figure 1 presents the results obtained when error terms are generated from a *t*-distribution (Case 1). All methods show an increased power to detect differential taxa as the sample size grows. The Huber, Bi_square, and LinDA90 methods perform comparably for small sample sizes, each showing higher power than the LinDA97 and LinDA methods. However, as the sample size grows, the performances of the LinDA90, LinDA97, Huber, and Bi_square methods converge, all surpassing the power of the LinDA method. Notably, the QR method consistently exhibits the lowest power and suffers from significant FDR inflation.

The results for error terms generated from a Log-normal distribution (Case 2) are presented in Figure 2. Like the previous case, all methods exhibit an increased power to detect differential taxa with a larger sample size. In both small and large sample size scenarios, the Huber and Bi_square methods lead in power. LinDA90 follows, outperforming LinDA97. Notably, when the sample size is small, LinDA90 has an FDR inflation. Upon closer inspection, the Huber method outperforms the Bi_square method regarding power and FDR control. Although the QR method has higher power than LinDA when n=200, it comes at the cost of considerable FDR inflation.

The outcomes for the setting in which error terms are sampled from a Weibull distribution (Case 3) are shown in Figure 3. As the sample size increases, all methods exhibit enhanced power in detecting differential taxa. When the sample size is small, all methods display limited power for detecting differential taxa, with a power of approximately 0.1. In contrast, with larger sample sizes, the Huber method outperforms all other methods, followed by the Bi_square and LinDA97 methods. Notably, Bi_square shows FDR inflation when the sample size is small, whereas the LinDA90 method exhibits significant FDR inflation with large sample sizes. Since the FDR of the QR method exceeds 0.3 when n=200, it was excluded from the plot.

#### 3.1.2. With Outliers Setting

We employ the following procedure to generate the data to demonstrate the impact of the number of outliers on the power of the DAA method. The number of taxa is fixed at m=500, and the number of samples is set to n=100. Following the data-generation process discussed in Section 3.1, we begin by sampling error terms from a normal distribution with a mean of 0 and a variance of $\sigma_{i*}^{2}$ for the *i*-th taxon. The outliers are generated by randomly selecting a subset of nonzero counts and multiplying them by a fold change of 20. More specifically, we randomly choose 250, 500, 1000, and 2000 nonzero counts, corresponding to an average of 0.5, 1, 2, and 4 outliers per taxon, respectively.

Let ρ denote the average number of outliers per taxon. The results for the case when outliers exist without confounders are presented in Figure 4. As the number of outliers increases, the power of all methods will decrease. When the number of outliers is small (ρ=0.05), LinDA97 exhibits slightly better performance across all methods. However, as the number of outliers increases, the performance of Huber and LinDA90 improves. Specifically, when ρ=4, LinDA90 demonstrates the highest power for small signal strengths, while Huber exhibits the highest power for large signal strengths.

## 4. Real Data Analysis

We utilized three real datasets, including independent samples from studies on *C. difficile* infection (CDI) [30], inflammatory bowel disease (IBD) [31], and rheumatoid arthritis (RA) [12]. The CDI and RA datasets were downloaded from the links provided in the original paper, while the IBD dataset was obtained from the Qiita database [32] using study IDs 1460 and 524. To ensure data quality, we excluded samples with less than 1000 read counts and taxa that appeared in less than 10% of the samples for each dataset. The variable that we are interested in is binary phenotypes across all datasets. In the case of the IBD dataset, the confounding factor is the usage of antibiotics.

First, we compared the detection power of all methods discussed in Section 3 across the three datasets. Figure 5 presents the number of discoveries at various FDR levels (0.01–0.25). For the CDI dataset, the LinDA97, LinDA, and Huber methods exhibited the highest number of discoveries at an FDR level of 0.1. As the FDR level increased, the Bi_square method identified the most discoveries. For the IBD dataset, all methods except QR had a similar number of discoveries at an FDR level of 0.1. However, the Bi_square and Huber methods outperformed others regarding discoveries at different FDR levels in the overall analysis. Conversely, the LinDA and Huber methods consistently identified the most discoveries across all FDR levels in the RA dataset. Therefore, the Huber method displayed superior discovery capability overall. It is worth mentioning that the QR method consistently yielded the fewest discoveries in all three datasets, which is consistent with our simulation results.

Subsequently, to illustrate the overlapping discoveries among different methods, we employed the UpSet plot [33] to depict the overlap at the target FDR level of 0.1. Figure 6 presents the overlap of differentially abundant taxa across the three real datasets at this FDR level. It is evident that, in most cases, the taxa identified by the Huber method are consistently identified by other methods as well. This consistency implies that the taxa identified by the Huber method are more likely to be “truly” differentially abundant. Conversely, the remaining methods exhibit independent findings, lacking support from other methods, suggesting a higher likelihood of false discoveries. Consequently, the Huber method demonstrates greater robustness in practical applications.

As an illustration, we present a boxplot of the taxa expression after applying the CLR transformation, focusing solely on taxa identified only by the LinDA method. Figure 7 displays an example of a taxa expression after CLR, identified only by the LinDA method in the RA dataset. The left panel shows the original data, while the right panel represents the data with potential outliers removed. In this case, we remove samples whose taxa expression after CLR exceeds 1.5. After removing these samples that significantly deviate from the rest, we observe similar means between the two groups. However, LinDA identifies this taxon as differentially abundant due to outliers, which may be a false discovery.

## 5. Conclusions and Discussion

This study investigates the influence of heavy-tailedness and outliers on differential abundance analysis using different methods and proposes effective strategies to address them. The presence of heavy-tailedness and outliers can substantially reduce the power of existing DAA methods. To resolve this issue, we propose two techniques to enhance the robustness of DAA methods. First, we introduce a general regression framework for DAA by extending the LinDA method. This framework includes differential analysis based on Huber regression as a special case, which is more stable in handling outliers. Second, we propose the winsorization method, which involves winsorizing the data before applying DAA methods.

Our findings from both simulation and real data indicate that, among the M-estimation-based approaches, the Huber method, in particular, demonstrates more robust performance in handling heavy-tailedness and the presence of outliers in the dataset. Specifically, compared to LinDA with winsorization, the Huber method exhibits higher power while maintaining FDR control. Furthermore, the Huber method demonstrates better FDR control than the Bi_square method. Consequently, we recommend utilizing the Huber method as an effective approach to address outliers and heavy-tailedness. Additionally, the optimal quantile level for winsorization remains unclear. Therefore, another advantage of the Huber method is that it does not require the selection of such hyperparameters.

## Figures and Tables

**Figure 1 genes-14-02000-f001:**
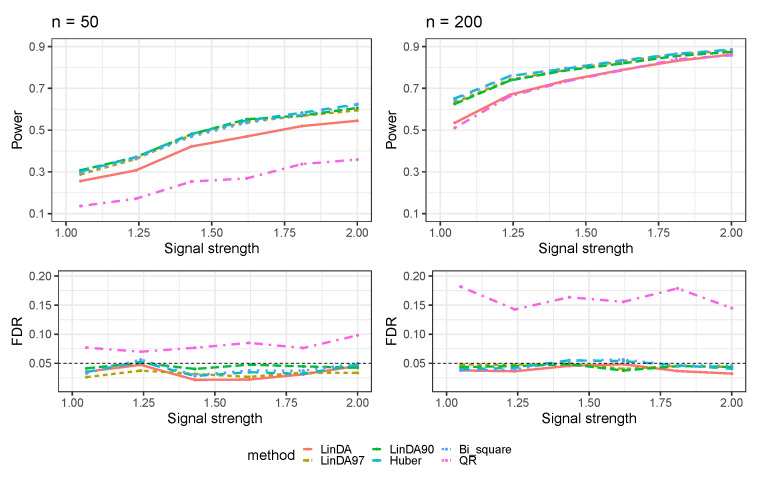
The log-linear model yielded results in the absence of confounding variables in the sparse-signal setting ($p_{\gamma }=0.05$), where errors were generated from a *t*-distribution. The left panel represents a sample size of n=50 and the right panel corresponds to a sample size of n=200.

**Figure 2 genes-14-02000-f002:**
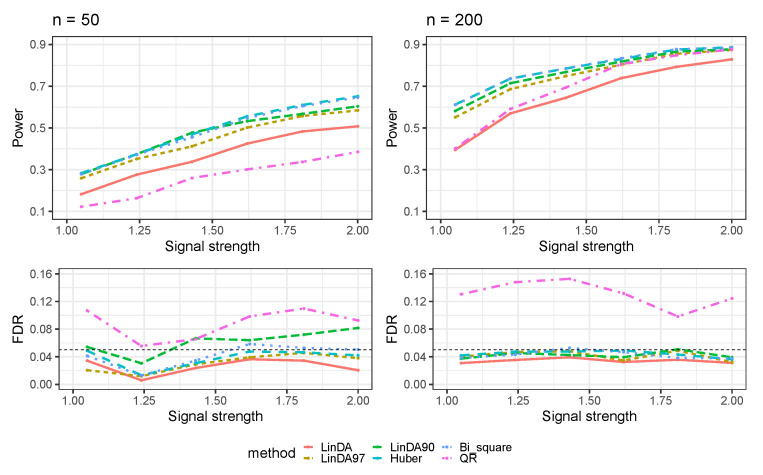
The log-linear model yielded results in the absence of confounding variables in the sparse-signal setting ($p_{\gamma }=0.05$), where errors were generated from a Log-normal distribution. The left panel represents a sample size of n=50 and the right panel corresponds to a sample size of n=200.

**Figure 3 genes-14-02000-f003:**
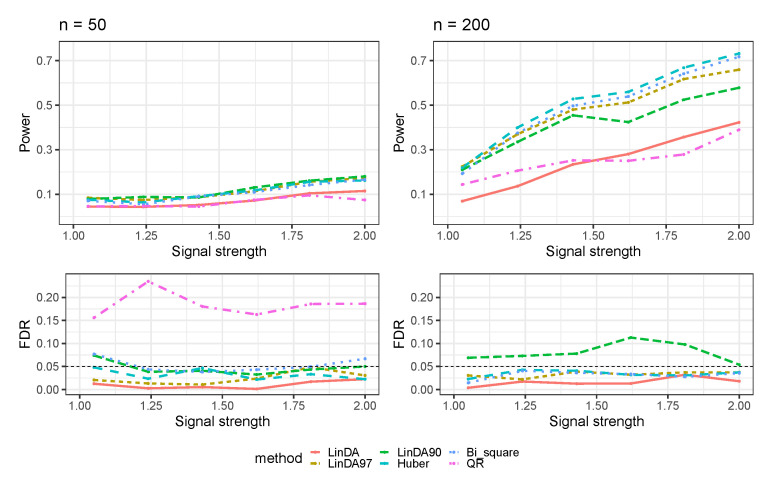
The log-linear model yielded results in the absence of confounding variables in the sparse-signal setting ($p_{\gamma }=0.05$), where errors were generated from a Weibull distribution. The left panel represents a sample size of n=50 and the right panel corresponds to a sample size of n=200.

**Figure 4 genes-14-02000-f004:**
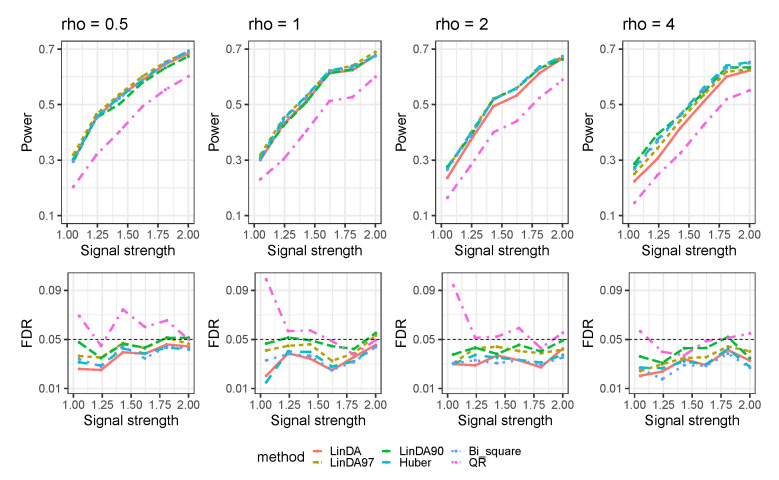
The log-linear model yielded results in the absence of confounding variables in the sparse-signal setting ($p_{\gamma }=0.05$) with outliers. ρ is the average number of outliers per taxon.

**Figure 5 genes-14-02000-f005:**
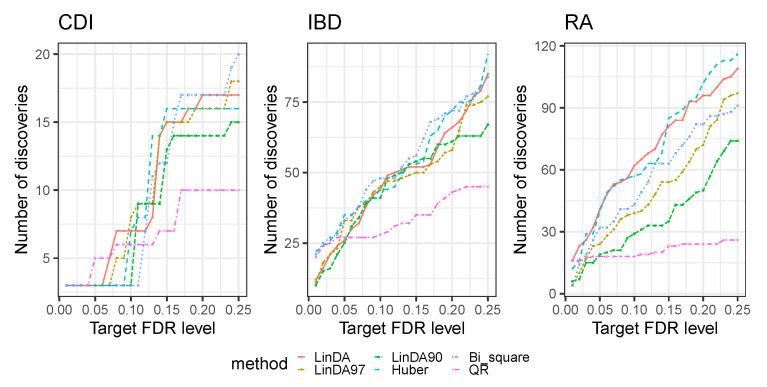
Number of discoveries with respect to different FDR levels (0.01–0.25) for three real datasets.

**Figure 6 genes-14-02000-f006:**
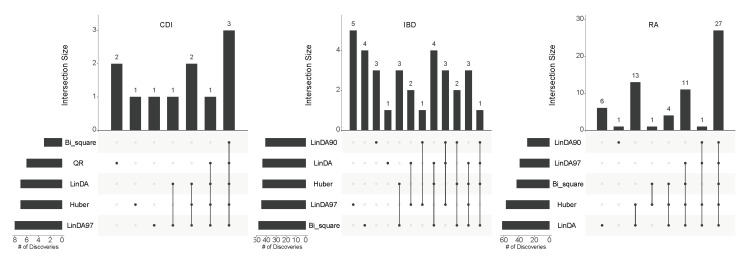
The overlap of differentially abundant taxa detected by various DAA methods across three real datasets at an FDR level of 0.1.

**Figure 7 genes-14-02000-f007:**
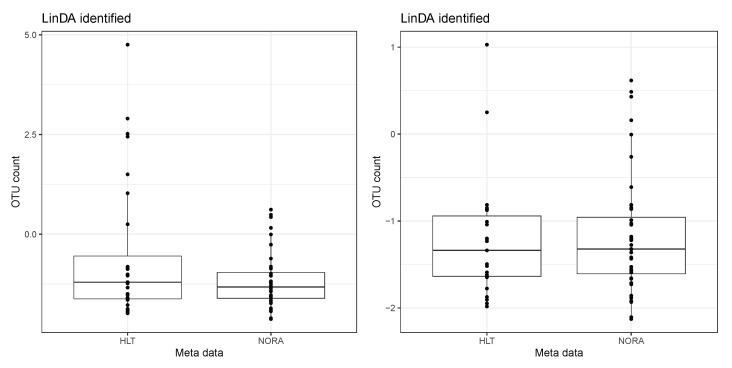
An example of a taxa expression after CLR. The taxa are only identified by the LinDA method in the RA dataset. The left panel shows the original data. The right panel represents the data with potential outliers removed.

## Data Availability

The function for implementing Robust DAA is available at https://github.com/guanxunli/robustDAA (accessed on 24 June 2023). The code for running the simulation is available at https://github.com/guanxunli/robust_daa (accessed on 23 October 2023).

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
