# Peer review of "Robust Differential Abundance Analysis of Microbiome Sequencing Data"

_genes, 2023, doi:10.3390/genes14112000_

Round 1

Reviewer 1 Report

Comments and Suggestions for Authors

This paper examines the impact of outliers and heavy-tailedness on DAA methods and investigate different strategies to mitigate it. Using LinDA as the benchmark DAA method, the authors identify the most effective strategy to address outliers and heavy-tailedness in DAA. Simulation studies demonstrate that the presence of outliers or heavy-tailedness significantly reduces the power in detecting differential taxa. To address the challenges of outliers and heavy-tailedness, the authors generalize the LinDA method by introducing a general M-estimation framework for DAA. This framework includes differential analysis based on Huber regression as a special case. Simulations reveal that Huber regression demonstrates superior robustness against both outliers and heavy-tailedness when compared to the LinDA method and the LinDA method with winsorization.

The topic of this paper follows the scope of the journal. Before I can give a positive comment, the authors should address the following issues:

1.      In the abstract, the authors should provide the full name of LinDA.

2.      It is suggested to list the main contributions of this paper at the end of the introduction part.

3.      In line 84, under (1), the authors give equation (2), it seems that there are no correlation between (1) and (2)

4.      As we know, the abundance of taxon may be highly nonlinear, some discussions on why linear models are used should be added.

5.      It seems that the main contributions of this paper are the comparisons among different methods, and thus the novelty of this paper is limited.

Author Response

We are very grateful for your feedback. We provide a detailed response to each of the comments in the attached pdf file.

Reviewer 2 Report

Comments and Suggestions for Authors

Robust differential abundance analysis of microbiome sequencing data

Summary:

This paper aims at studying the impact of outliers and heavy distribution tails on differential abundance analysis from microbiome sequencing data. The authors use a differential abundance analysis tool named LinDA and that they previously developed.

Major comments:

-       While I understand that the authors might want to focus on the tool they previously developed, it is crucial that the question of interest includes other independent tools. For a benchmark study to be complete, it needs to include non-overlapping methodologies. This would also help in generalizing the study findings and provide more insight as to how this method might improve the outputs of other differential abundance analysis methods.

-       The Huber method is suggested as a strategy for noisy data. For users to be able to employ this method, it would be important to define noise in the context of this study. Moreover, what would be the impact of using this method on non-noisy data?

Minor comments:

-       The impact of outliers and heavy distribution tails on statistical power is described as a component of the presented study however, in the abstract, it is unclear if this was studied previously or as part of this study, and this should be clarified.

-       In the introduction, the authors discuss absolute measurements of microbial abundance and suggests that estimating absolute values is not feasible. However, measuring input mass as well as including control material of known mass provides a valid way of estimating those values.

-       In Figure 7, the boxplots should present individual data points.

Author Response

(The authors gave the same response as above.)

Round 2

Reviewer 2 Report

Comments and Suggestions for Authors

I thank the authors for the detailed answers to my comments. I believe the edits made contribute to building a stronger case.